# Learning fast and fine-grained detection of amyloid neuropathologies from coarse-grained expert labels

Daniel R. Wong [1,2,3,4,5], Shino D. Magaki[6], Harry V. Vinters[6,7], William H. Yong[8], Edwin S. Monuki[8], Christopher K. Williams[6], Alessandra C. Martini [8], Charles DeCarli [9], Chris Khacherian[8], John P. Graff [10], Brittany N. Dugger [10✉] & Michael J. Keiser [1,2,3,4,5✉]

Precise, scalable, and quantitative evaluation of whole slide images is crucial in neuropathology. We release a deep learning model for rapid object detection and precise information on the identification, locality, and counts of cored plaques and cerebral amyloid angiopathy (CAA). We trained this object detector using a repurposed image-tile dataset without any human-drawn bounding boxes. We evaluated the detector on a new manually-annotated dataset of whole slide images (WSIs) from three institutions, four staining procedures, and four human experts. The detector matched the cohort of neuropathology experts, achieving 0.64 (model) vs. 0.64 (cohort) average precision (AP) for cored plaques and 0.75 vs. 0.51 AP for CAAs at a 0.5 IOU threshold. It provided count and locality predictions that approximately correlated with gold-standard human CERAD-like WSI scoring ($p = 0.07 \pm 0.10$). The openly-available model can quickly score WSIs in minutes without a GPU on a standard workstation.

[1] Institute for Neurodegenerative Diseases, University of California, San Francisco, San Francisco, CA 94158, USA. [2] Bakar Computational Health Sciences Institute, University of California, San Francisco, CA 94158, USA. [3] Department of Pharmaceutical Chemistry, University of California, San Francisco, San Francisco, CA 94158, USA. [4] Department of Bioengineering and Therapeutic Sciences, University of California, San Francisco, San Francisco, CA 94158, USA. [5] Kavli Institute for Fundamental Neuroscience, University of California, San Francisco, San Francisco, CA 94158, USA. [6] Department of Pathology and Laboratory Medicine, University of California, Los Angeles, Los Angeles, CA 90095, USA. [7] Department of Neurology, David Geffen School of Medicine at University of California, Los Angeles, Los Angeles, CA 90095, USA. [8] Department of Pathology & Laboratory Medicine, University of California, Irvine, CA 92697, USA. [9] Department of Neurology, School of Medicine, University of California-Davis, Davis, CA 95817, USA. [10] Department of Pathology and Laboratory Medicine, School of Medicine, University of California, Davis, Sacramento, CA 95817, USA. ✉email: bndugger@ucdavis.edu; keiser@keiserlab.org

Deep phenotyping of Alzheimer disease requires accurate evaluation of whole slide image (WSI) data[1]. For amyloid-β (Aβ) pathologies, such as plaques and cerebral amyloid angiopathy (CAA), quantifying Aβ burden phenotypes in the brain can aid in understanding disease mechanisms and progression[2–4]. There is a great need for quantitative, scalable means of assessing neuropathologies as current practices can suffer from interrater reliability issues and limited statistical analysis power, especially when relying on semi-quantitative scores. Current neuropathology assessments for AD pathologies such as NFT stage[5,6] and amyloid plaque phase[7] have added immensely to understanding the pathological progression of neurodegenerative diseases. Most schematics have used ordinal scales or the presence or absence of pathologies in specific areas. Recently, quantitative neuroanatomic-specific data have aided more robust correlations and understanding of selective vulnerability. This has led to further precision medicine approaches and the emergence of disease subtypes[1,8–10]. However, in neuropathological practice, quantifying Aβ burden has primarily been semi-quantitative[11] with variable interpretation[12,13]. Interpreting WSIs is a time-consuming task[14] with experts regularly spending many hours a day assessing slides[15].

Deep learning has helped to address these challenges, providing quantitative and automated solutions to identifying and quantifying Aβ burden[13,16,17]. Deep learning can augment neuropathologist expertise[16] and combine multiple expert annotations into a robust and automated labeler[13]. For more localized tasks like object detection[18] and semantic segmentation[19], deep learning has also provided accurate and automated means of quantifying Aβ and tau neuropathologies[20–22]. However, such studies require significant human expert labor to create high-quality training datasets in the form of manually drawn bounding boxes or segmentations and categorical labels. Furthermore, the models typically require specialized dedicated and expensive hardware like graphic processing units (GPUs)[23], without which the prediction task of quantifying pathologies can take hours for even a single WSI. Furthermore, as with many deep learning studies, generalizability to data from different institutions is difficult to guarantee[17,24,25].

Here, we present a fast You Only Look Once version three (YOLOv3) based model[26] that rivals human-expert level detection of cored plaque and CAA pathologies. Moreover, we created this model from a dataset not intended for object detection, requiring much less human labor than a traditional object detection dataset. We evaluated the model on WSIs outside of its training corpus, which were diverse in both stain and institutional source. The model, released at https://github.com/keiserlab/amyloid-yolo-paper, can quickly score WSIs without a GPU, paving the way for more accessible and equitable deep learning applications in the research and clinical space. Furthermore, we showed that without a GPU, the model can still score WSIs in a matter of minutes, with speed improvements of at least eight times over various state-of-the-art deep learning approaches for quantifying neuropathologies[16,22]. To determine the model's potential for adoption of widespread scoring use, we evaluated it on WSIs with known CERAD-like scores[11,27] and found strong correspondence. The model enables scalable, reproducible, and precise detection for rapid clinical research applications.

## Results

### We built an object detector from a noisy and sparse dataset.
We repurposed a publicly available dataset from a previous study[13] not meant for object detection training. The previous study had collected human annotations post hoc on a 256 × 256 pixel tile basis for tiles centered on approximate bounding boxes

of cored plaque and CAA pathologies derived from traditional and automated computer vision techniques (see "Methods"). Consolidating this dataset into 659 larger field-of-view (1536 × 1536 pixel) images devoid of human-drawn boxes, we reformatted the data to a form more suitable for object detection. This dataset had many limitations: (1) a single pathology often incorrectly spanned many approximate boxes, particularly for CAAs (Supplementary Fig. 1); (2) the 1536 × 1536 fields lacked comprehensive annotations, resulting in a sparse label set prone to false negatives; (3) traditional watershed techniques defined each box, rather than human intelligence; (4) the dataset size was relatively small (659 images from 29 WSIs); and (5) due to limitations 1, 2, and 3 there was no reliable quantitative benchmark to assess model performance. For this study, we did not collect further human annotations for training, instead adapting the existing dataset to a more suitable form. Limitations 2–5 would have required additional human annotation work. To solve limitation 1, we performed an iterative merging procedure such that overlapping label boxes of the same class were joined (Supplementary Fig. 1; "Methods").

Once we merged the label data for use in the current study, we trained a YOLOv3 network[26] to identify cored plaque and CAA pathologies ("Methods"). We denote this initial model as model-1. Average precision (AP) for each class at varying intersection-over-union (IOU) thresholds typically exceeded 0.6 (Supplementary Fig. 2a); however, the model had some flaws. Visually, model-1 incorrectly labeled single instances with many overlapping boxes, especially for CAAs (Supplementary Fig. 2b). Hence, we trained a new model incorporating two enhancements. For the first, we joined overlapped output CAA predictions from model-1 to consolidate the fragments ("Methods"). Secondly, we used our previously released consensus-of-two convolutional neural network (CNN) model[13] to filter out low-quality CAA detections by removing detections that did not meet a final CNN classification prediction of 0.5 or higher. Finally, we merged output boxes of the same class, arriving at our final model, denoted model-2 ("Methods"). Figure 1a shows model-2's average precision over varying IOU thresholds (Supplementary Fig. 3) and example image predictions (Fig. 1b). Although model-2's average precision over varying IOU thresholds (Fig. 1a) does not greatly differ from model-1's, model-2 identified pathologies better by visual evaluation. This is sensible, as improved bounding box quality not only improved training but also increased the stringency of the validation benchmark. Consequently, we used model-2 for the remainder of the study.

### Fine-grained human expert annotations of pathologies were variable.
To assess the model's prospective capabilities, we needed a higher quality test dataset (one free from limitations 1-3) to derive reliable quantitative metrics. For this, four experts (anonymized as NP1-NP4) independently annotated an entirely new dataset from a new decedent cohort, drawing boxes around and classifying pathologies ("Methods"). These test data differed markedly from our training and validation data, which only had sparse and incomplete computer-generated boxes with expert labels. This new dataset consisted of 200 1536 × 1536 pixel images spanning four different immunohistochemical stains for Aβ deposits. We found that the four neuropathology annotators did not always agree on this fine-grained task, with average agreement accuracy for cored = $0.43 \pm 0.05$ and CAA = $0.33 \pm 0.11$ at an IOU threshold = 0.50 (Fig. 2a). Given this variability, we additionally created a "consensus annotation" benchmark set wherein each "positive" object and its box were independently supported by at least two out of the four annotators (Fig. 2b; "Methods").

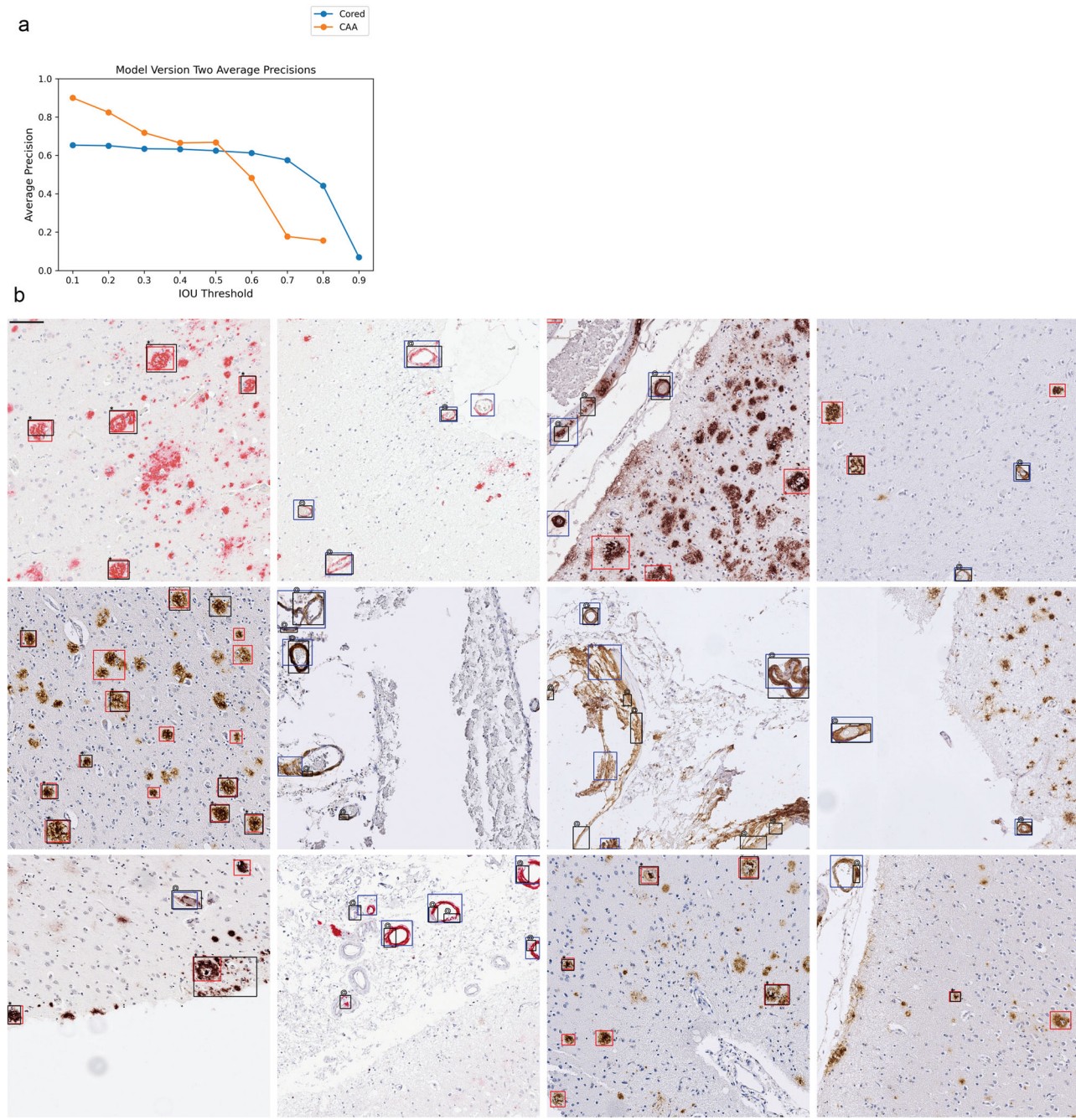

**Fig. 1 Model-2 performance and example image predictions.** Cored = Cored plaque, CAA = cerebral amyloid angiopathy. **a** Average precisions (AP) over the validation set at various IOU thresholds. The AP at IOU = 0.90 is undefined for CAA. Positive annotation sample sizes for Cored = 1274, CAA = 355. **b** 12 example images from the validation set. Cored prediction: red, cored label: black "*"; CAA prediction: blue, CAA label: black "@". Note that these training label data are sparse and do not contain every pathology ("Methods"). Stains first row (left to right): 6E10, 6E10, NAB228, 4G8. Second row: 4G8, 4G8, 4G8, 4G8, third row: NAB228, 6E10, 4G8, 4G8. Scale bar = 100 μM at upper left.

**The model achieved human-expert-level precision.** When we assessed the model against both individual-expert annotations and the consensus annotation datasets, we found it achieved expert-level precision at identifying both cored plaque and CAA pathologies on these new datasets despite never having been trained on manual bounding boxes (Fig. 3a). To cross-compare expert consistency across the annotations and thereby determine the achievable performance range from human variability, we treated each expert's annotations as though they were model predictions and compared them against each other. In this procedure, each annotator's labels sequentially became a ground

truth benchmark, against which we compared every other expert's annotations; we subsequently calculated the average precision (shown as the blue-dotted line in Fig. 3a). For most IOU thresholds (less than or equal to 0.70) and most benchmarks, the model operated within the range of human-expert level performance. For CAAs, the model's AP exceeded the average AP between human experts for four out of five benchmarks at IOU thresholds less than or equal to 0.60. For the strictest IOU thresholds ≥0.80, which require a more exact match between the predicted and label bounding box coordinates, the model fell short of human-expert performance (which itself was relatively

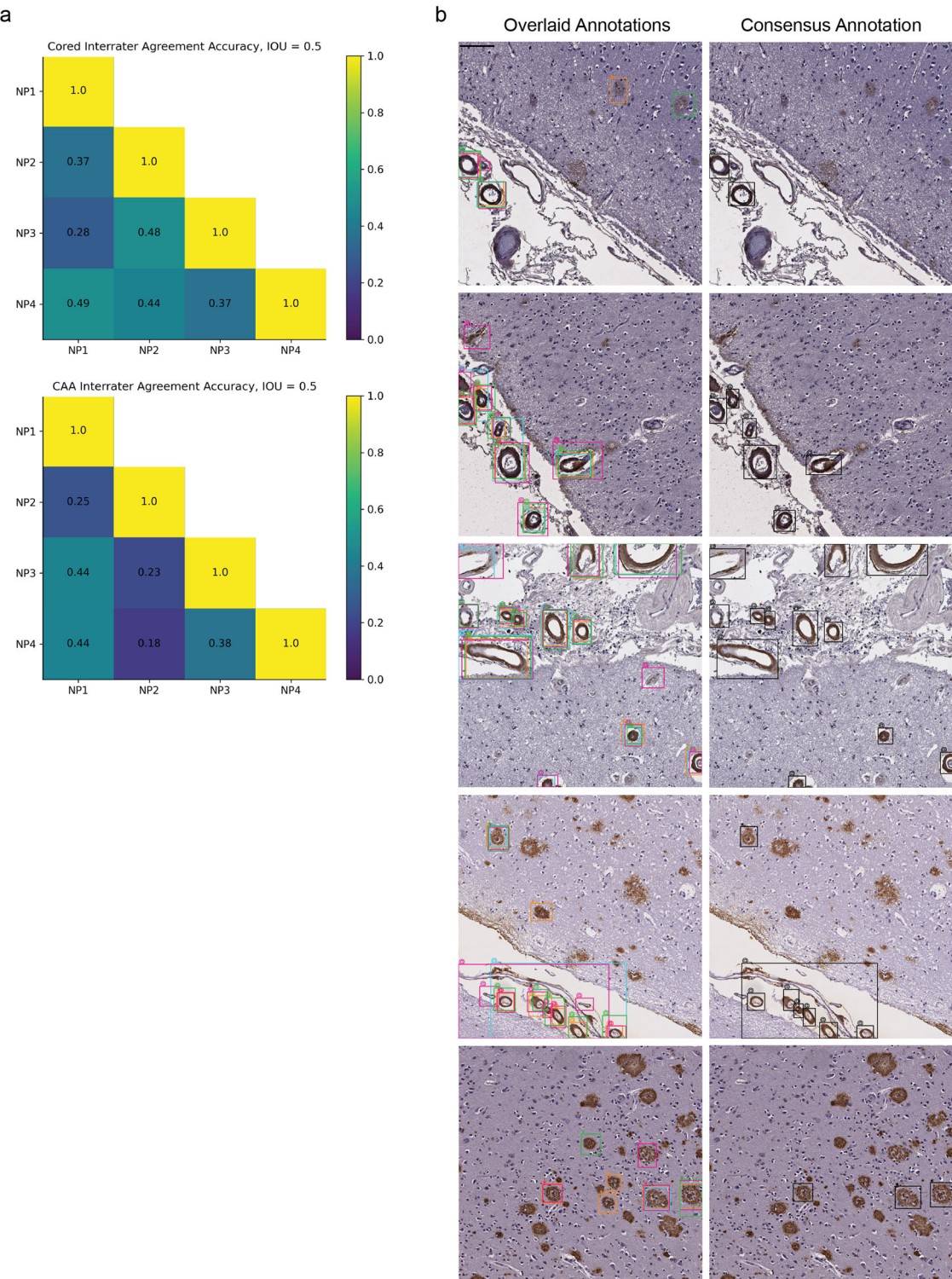

**Fig. 2 Fine-grained human bounding-box style annotations vary slightly. a** Interrater agreement accuracy among annotators, with a minimal IOU threshold of 0.50 used for counting two objects of the same class as an overlap ("Methods"). **b** Left column: example overlaid annotations from each of the four annotators (each a different color); right column: corresponding consensus annotation. Scale bar = 100 μM.

low) for four out of five benchmarks. Of all the human experts, the model's predictions most closely matched the annotations of NP1, who spent significantly more time annotating than any of the other annotators (Fig. 3b). NP1 spent nearly three times as long as NP2 and about twice as long as NP3 and NP4. Visually, the model predictions closely matched consensus annotations

(Fig. 3c). Model-1 also achieved human-expert-level performance (Supplementary Fig. 4), although model-2 outperformed it.

**Model predictions correlated with clinical CERAD-like scores**. We sought to test whether the model could quantify select Aβ

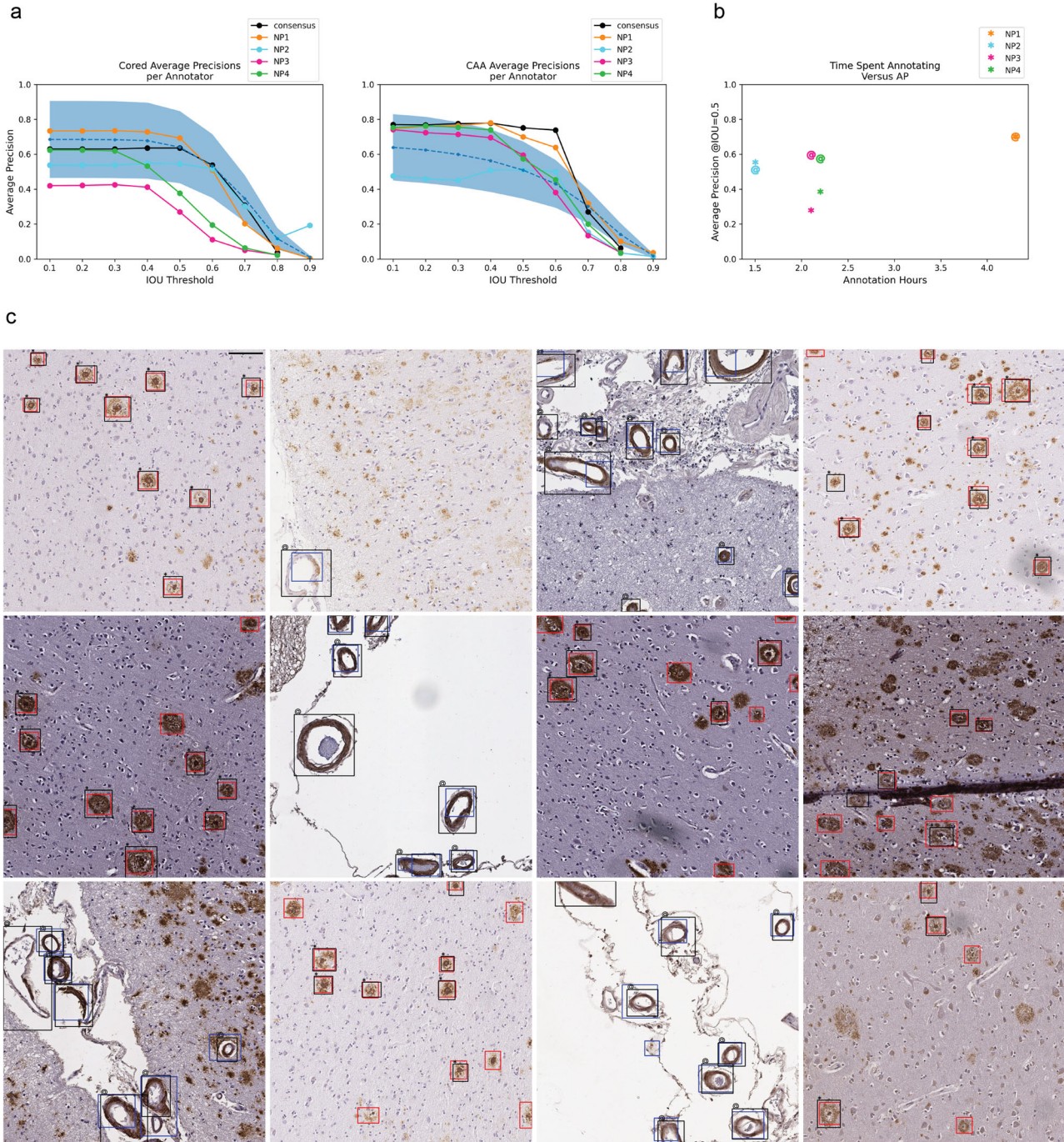

**Fig. 3 Model achieved human-expert level performance at identifying cored and CAA pathologies. a** Average model precision scores for identifying cored plaque (left) and CAA pathologies (middle). Y-axis: average precision, x-axis: IOU threshold that determines the minimal IOU required for a prediction to overlap with a label to be a true positive. Higher IOU thresholds are more stringent. The figure legend indicates which of the annotators is the ground truth benchmark for assessing the model (NP1 = neuropathologist 1, NP2 = neuropathologist 2, etc). The black line indicates model AP against the consensus annotator benchmark (Fig. 2b, right column). The blue dotted line is the average precision of comparing expert annotators to each other ("Methods"). The blue-shaded region is one standard deviation above and below the average-expert precision. Sample sizes (ground truth, Cored positive annotations, CAA positive annotations): (NP1, 370, 259), (NP2, 153, 153), (NP3, 121, 395), (NP4, 235, 324), (consensus with IOU = 0.50, 231, 289). **b** Total hours each annotator spent annotating (x-axis) versus AP at IOU = 0.50 of the model on the annotator's benchmark (y-axis). "*" indicates cored plaque performance, "@" indicates CAA performance. **c** Model predictions overlaid against consensus annotation. Cored plaque prediction: red, cored plaque label: black "*"; CAA prediction: blue, CAA label: black "@". The consensus annotation defines the labels. Scale bar = 100 µM.

deposits, CAAs, and cored plaques on WSIs. Hence, we asked if an automated score calculated for entire WSIs based on the model's detection of Aβ pathologies would reflect human-expert-based CERAD-like category scoring[11]. We used a testing holdout set of 63 WSIs labeled with this semi-quantitative gold standard for pathology from a previous study[16].

We used our model to exhaustively detect and count pathologies within each of the 63 WSIs. We found that the

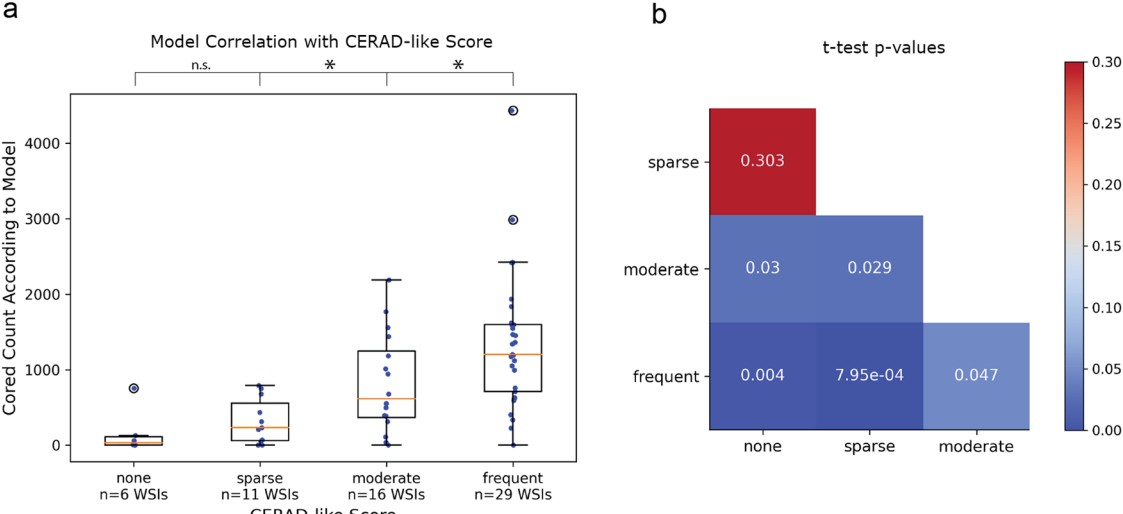

**Fig. 4 Model correlated with clinical CERAD-like scoring. a** Box plots for each CERAD-like category. Y-axis is the model-derived count of cored plaques, and the x-axis is the CERAD-like category. Scatter plot overlaid as blue dots (each dot corresponds to a unique WSI). Hollow black circles indicate outliers outside the third quartile plus 1.5x interquartile range. *$p < 0.05$, n.s. is not statistically significant. **b** P-values from a two-sided student's t-test comparing model-derived count distributions between each CERAD-like category.

counts (Fig. 4a) derived from the model predictions significantly correlated with CERAD-like severities (Fig. 4b). We performed a two-sided student's t-test to determine if the model-derived count distributions differed significantly between the different CERAD-like categories. We found significant differences for all category pairs at an alpha of 0.05, except for the pair "none" and "sparse" (student's t-test $p$-value = 0.30, power = 0.62). Power for all other comparisons exceeded 0.99.

**The model is much faster than existing approaches**. Next, we evaluated the model's practical usability as measured by its speed. Hence, we evaluated the 63 WSIs from our CERAD-like dataset using one NVIDIA Titan Xp GPU and computed the model's average speed per WSI. The model averaged one minute and forty seconds to score a single WSI. Consumer-grade GPUs are not universally available in pathology practices, so we tested model speed without GPU; the model averaged five minutes and thirty seconds per WSI on an Intel Xeon CPU (Supplementary Table 1).

We compared this YOLOv3 model's evaluation time with two different deep-learning approaches for quantifying neuropathology burden. First, we compared with our previously published approach of using a CNN sliding window to count plaque burden, which took three hours and four minutes per WSI on an NVIDIA GTX 1080 GPU[16]. Even without a GPU, this YOLOv3 model was 33 times faster (5 min 30 s/WSI) than the older GPU-*enabled* sliding window approach (181 min 30 s/WSI). On average, when we used a GPU, the YOLOv3 model was 110 times faster than our previous approach[28]. Second, we compared the YOLOv3 model's speed to a semantic segmentation method for tauopathies[22]. This different state-of-the-art approach to quantifying neuropathologies reported 45 minutes per WSI using a GPU; this YOLOv3 model was 8x-27x faster, depending on GPU usage. Exact runtimes will vary by hardware. It is important to note that GPU dependency or lack thereof is dependent on model architecture.

## Discussion
We present a rapid object-detection model for identifying Aβ pathologies, cored plaque and cerebral amyloid angiopathy across a range of immunohistochemically stained slides. Three points of

the study merit particular emphasis: (1) we developed a detector model from a dataset that did not require the same time and labor usually needed for building accurate object detectors; (2) the model matched human-expert performance; and (3) the model showed promise for usability without special GPU hardware for evaluation. Regarding the first point, we overcame one of the main problems with training object detectors through an iterative process: manual labeling and localization of high-quality bounding box data. The model still relied on accurate categorical labels in its training, but this is much less work than drawing a box and providing a categorical label, which is important for scalability. We hope our proof-of-concept encourages other deep learning studies to explore the prospect of bootstrapping from a more economical standpoint via more pragmatic proxy data, perhaps even by building more directly off of preliminary training data from conventional computer vision tools. We were encouraged to see the approach could identify the two pathologies with high precision, starting from only 659 initially noisy and sparsely-annotated high-resolution training images.

The final model achieved human-expert-level performance on a prospective test despite using a lower-quality training dataset devoid of human-derived bounding boxes. Although the study's scope of four expert annotators and 200 prospectively annotated images does not ensure generalizability to all experts, pathologies, stains, areas, cases, and institutions, it was encouraging to see the model most closely aligned with the annotator who spent the most time annotating (Fig. 3b). With expert-level precision on held-out data, models such as these may readily be used as a secondary or preliminary labeler by neuropathologists, particularly to flag unusual cases. The current model's effectiveness will depend strongly on whether an expert needs strict down-to-the-pixel bounding-box overlap between the prediction and the actual pathology because the model falls slightly below human-expert level performance at the strictest IOU thresholds. For annotation tasks demanding high locality, a pixel-by-pixel level of precision via semantic segmentation may be a more apt technical approach.

Accordingly, several caveats apply to the study. First, although our prospective test dataset was composed of cases across three different institutions, variable in stain, and annotated by four experts, it consisted of only two hundred $1536 \times 1536$ pixel images derived from 56 WSIs. There can be numerous variables

that alter algorithm outputs[29]. Therefore much of the model's generalizability has yet to be explored. Likewise, performance on stains outside of the four used has yet to be determined, although we did not see much performance variation by stain except for 6E10 (Supplementary Fig. 5). Furthermore, CAA pathology is diverse[30], but this model does not differentiate between subtypes; we would be interested to explore CAA-subtype identification in a future study. We confined our study to slides immunohisto-chemically stained with antibodies against Aβ. We have not determined the model's efficacy on other stain types, like hema-toxylin and eosin (H&E) or silver stains. Finally, we relied on author-reported runtime analyses for various comparative speed benchmarks against alternative computational methods when their code was unavailable. These assessments necessarily spanned 1–2 generations of CPU and GPU hardware. However, given a GPU is approximately two orders of magnitude faster at deep learning tasks than the contemporaneous CPU, we found the CPU-only speedup of the YOLOv3 model against GPU-enabled alternatives compelling.

The model's predicted Aβ deposit counts correlated sig-nificantly with CERAD-like category scoring at the WSI level (Fig. 4a) without any training specifically for this purpose. A score of predicted-object counts struggled to significantly differentiate the "sparse" versus "none" CERAD-like categories, perhaps due to the low sample sizes ($n = 6$ and $n = 11$) and resulting in a low power of 0.62. Nonetheless, we hope the model can quickly assess WSIs and provide a proxy for CERAD-like scoring, especially in detecting cases with higher plaque burden.

We freely release the trained model, annotated dataset, and study source code for easy access and use. We provide an example environment (using conda) to standardize model deployment. Although we found the consensus-annotation benchmark seemed a more stable label dataset than any given individual expert's annotations *on average*, consistent with the common notion of the "wisdom of the crowd," we did not tune the model for *specific* expertise nor neuropathology focus areas. Consequently, inter-ested researchers may wish to fine-tune or entirely retrain this model on more precisely formulated annotations to fit their needs. We hope the model and dataset's open-source release, enabling quick evaluation of WSIs even without a GPU, will facilitate the shareable and scalable application of deep learning in neuropathology.

## Methods

**Training and validation dataset preparation**. The training of model-1 and model-2 builds on the methods and dataset first presented in Wong et al.[13]. We provide a brief description of the methods used to build this dataset. We collected 29 WSI brain sections of the temporal cortex from 3 different sites: 11 from the Alzheimer Disease Research Center at the University of California, Davis (UC Davis); 11 from the University of Pittsburgh; and seven from UT Southwestern (Supplementary Fig. 6). Slides were derived from formalin-fixed paraffin-embed-ded sections and stained with antibodies directed against Aβ. UC Davis used the Aβ 4G8 antibody (1:1600; Covance Labs, Madison, WI, USA), the University of Pittsburgh used a NAB228 antibody, and UT Southwestern used a 6E10 antibody. UC Davis used the chromogen 3,3′-diaminobenzidine and counterstained with hematoxylin. University of Pittsburgh used the chromogen Nova Red from Vector (Sk-4800). UT Southwestern used the Leica Bond robotic immunostaining plat-form with proprietary detection reagents. The detection kit used a red chromogen which employs an alkaline phosphatase enzyme and a Fast Red chromogenic substrate. We imaged all WSIs on an Aperio AT2 at either ×20 or ×40 magnifi-cation at the different institutions. We resized all ×40 images to ×20. For partici-pant demographic data, please refer to Wong et al.[13].

We color-normalized the WSIs[31]. Each WSI was uniformly tiled to 1536 × 1536 pixel non-overlapping images. After tiling, we applied a hue saturation value (HSV) color filter and smoothing technique to detect candidate plaques using the python library openCV. We used different HSV ranges for the different stain types as follows: 4G8 HSV = (0, 40), (10, 255), (0, 220); NAB228 HSV = (0, 100), (1, 255), (0, 250); and 6E10 HSV = (0, 40), (10, 255), (0, 220).

Within each tiled 1536 × 1536 pixel image, each candidate pathology was bounding boxed via the watershed algorithm and then annotated by four

neuropathology experts. Each expert performed a multi-class labeling task, selecting any or none of the classes: cored plaques, diffuse plaques, and CAA. After annotation, we applied a consensus-of-two strategy to obtain our final label set, such that a candidate plaque $p$ was recorded as positive if any two experts marked $p$ as positive for class $c$, else we recorded $p$ as negative. For example, a consensus-of-two strategy means at least two annotators identified the pathology, even if the remaining annotators did not. We discarded the diffuse plaque pathologies from our dataset and focused on the classes cored plaques and CAA. Diffuse plaques are not well-defined as objects (hence the term diffuse) and are thus out of scope for an object-detection method. If any bounding box of class $c$ overlapped with any other bounding box of class $c$, we merged the boxes into a single bounding box of class $c$, which was the minimal superset of the two bounding boxes. The result was 659 1536 × 1536 pixel images that contained either a cored or CAA pathology.

**Training model-1**. We split the 29-WSI dataset of 659 images into 70% training and 30% validation. We trained an initial YOLOv3 network from the image dataset for 200 epochs using a pre-trained Darknet, a batch size of eight, and a learning rate of 0.001. The file config/yolov3-custom.cfg at https://github.com/keiserlab/amyloid-yolo-paper contains full training hyperparameters. We selected the model weights from the epoch giving us the highest mean average precision over the validation set.

**Training model-2**. We ran model-1 over the training set. We merged model output prediction boxes of the same class using the same data preprocessing procedure as the original bounding box merging. We combined the resulting merged predictions with the existing training labels to create a new training dataset. We then used the model published in Wong et al.[13]. to remove bounding boxes with predicted confidence <0.5 for the relevant deposit to filter unlikely (false-positive) model-1 predictions. We trained a new model from this new training dataset, called model-2. The training parameters were the same as those used for training model-1.

**Selecting images for prospective test**. To assess our model on data it had never seen, we collected a new dataset of 56 WSIs that differed from those used for training and validation. These new WSIs came from three different institutions: UC Davis, UC Los Angeles (UCLA), and UC Irvine (UCI). UC Davis used a 4G8 stain, UCLA used both an ABeta40 and ABeta42 stain, and UCI used a 6E10 stain (Supplementary Data 1). We imaged all WSIs on an Aperio AT2 (UC Davis), Aperio CS2 (UCLA), and an Aperio Versa 200 (UCI) scanner at ×20 or ×40 magnification at the different institutions. For UCLA and UC Davis slides, each pixel corresponds to 0.5 microns. For UCI slides, each pixel corresponds to 0.274 microns. For each of the four stains, we selected the top 12 WSIs with the highest count of human-annotated CAAs, resulting in 48 different WSIs. UC Davis and UCLA had an approximate tissue thickness for each WSI of 6 μm, while UCI had a tissue thickness of approximately 5 μm. UC Davis, UCI, and UCLA all used the chromogen 3,3′-diaminobenzidine.

For each of these 48 slides, we selected three 1536 × 1536 pixel fields to be used for prospective testing as follows: (1) field with the largest count of CAA positive model predictions (from model-2); (2) field with the largest count of CAA positive human annotations; and (3) top two fields with the largest count of cored positive model predictions (from model-2). In addition, for each of the four stains, we randomly selected two WSIs that were different from the original 48. We randomly picked two fields for each of these eight WSIs. This resulted in a total prospective test set size of 200 images, each with 1536 × 1536 pixels.

**Annotating prospective test images**. Four neuropathology experts independently annotated each of the 200 images used for prospective testing. These experts were different from the original five who helped to create our training and validation dataset in the prior study[13], except for one expert (B.N.D.) who helped with labeling the training and validation set and the prospective test images. We pro-vided standardized annotation instructions to all experts (Supplementary Note 1). We used the web platform called "SuperAnnotate"[32] for obtaining bounding box labels. Each annotator had one month to complete the annotations.

**Assessing interrater agreement**. We evaluated the interrater agreement accuracy between two annotators (denoted as "A1" and "A2" in this section) for prospective testing as follows. For the superset of all pathologies of class P (either cored or CAA) identified by A1 or A2 (with cardinality "total"), we determined if both annotators gave congruous labels for each plaque. Two label boxes form a con-gruous pair if they share the same class label and intersect with IOU threshold of at least 0.50. We allowed each label pathology to be a part of at most one congruous pair (i.e., if multiple of A1's labels overlapped with a single label from A2, only one of A1's labels was part of the pair). We defined "overlaps" as the number of congruous pairs between A1 and A2. The final interrater accuracy between A1 and A2 derives from "overlaps" divided by "total" (Fig. 2).

**Assessing model performance on the prospective test images**. We used model-2 to assess performance on the prospective test images. For each of the 200

images, we derived final predictions by merging any predicted bounding boxes that overlapped with any others of the same class. For each CAA bounding box prediction, we center-cropped the box to derive a $256 \times 256$ pixel image and fed this image into a previously published CNN model (the consensus-of-two model first presented in Wong et al.[13]). If the consensus-of-two model gave a negative CAA prediction, then this prediction was removed.

For all model evaluations, if there were multiple detections for a single label, we counted the highest confidence detection as a true positive and the rest as false positives (keeping consistent with the precedent set forth by the PASCAL VOC challenge[33]).

To determine the ceiling performance we could expect from our model, we assessed how well each expert annotator matched the other experts (blue shaded region in Fig. 3a). For each expert annotator (denoted as "A" for this section), we first fixed A as the ground truth, compared the other annotators' labels (not including A) to A's ground truth, and derived the precision. We did not include the consensus "annotator" in this comparison. We averaged the resulting 12 different comparisons and precision scores for each IOU threshold and plotted the standard deviation around the average (Fig. 3a).

**Correlating model-derived plaque counts with CERAD-like scoring**. The original CERAD scoring[27] aimed to denote the densities of neuritic plaques per $1\,mm^2$ area, and stains used consisted of Thioflavin-S or silver. As we were evaluating different Aβ deposits on Aβ-stained tissue, we utilized the term "CERAD-like". For each of the 63 WSIs with CERAD-like scores available in Tang et al.[16], we tiled the WSI into non-overlapping $1536 \times 1536$ pixel tiles. We ran model-2 over all tiles to identify predicted pathologies. We merged any predicted bounding boxes that overlapped with any others of the same class and counted the number of predicted cored bounding boxes to compare with the clinical CERAD-like score.

We performed a two-sided student's t-test between each CERAD-like category's distribution of model-derived pathology counts. The null hypothesis was that the two distributions were no different, and the alternative hypothesis was that the two distributions were indeed different. Each point of any distribution was a single model-derived plaque count from one WSI. We used an alpha threshold of 0.05 to assign significance.

**Statistics and reproducibility**. For all hypothesis testing, we used a standard threshold of $p < 0.05$ to assign significance. All results are reproducible, with source code located at https://github.com/keiserlab/amyloid-yolo-paper. Where appropriate, sample sizes are defined in the Figures and Figure Legends.

**Ethics declarations**. Materials utilized for these studies consisted of human postmortem brain samples that were digitized into digital WSIs. Only living subjects are defined as Human Subjects under federal law (45 CFR 46, Protection of Human Subjects). All participants or a legal representative of the participant signed informed consent during the participant's life as part of the University of California Alzheimer's Disease Research Center programs. All human subject involvement was overseen and approved by the Institutional Review Board (IRB) at the relevant University of California site. All data followed current regulations, laws, and IRB guidelines.

**Reporting summary**. Further information on research design is available in the Nature Portfolio Reporting Summary linked to this article.

## Data availability
All image data is freely available at DOI: 10.17605/OSF.IO/FCPMW (https://doi.org/10.17605/OSF.IO/FCPMW).

## Code availability
The complete source code and fully trained models are available at: https://github.com/keiserlab/amyloid-yolo-paper[34] and https://doi.org/10.5281/zenodo.7944799.

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

## Acknowledgements

The authors thank the families and participants of Alzheimer's Disease Research Centers for their generous donations as well as ADRC staff and faculty for their contributions. This work was supported by grant number 2018-191905 from the Chan Zuckerberg Initiative DAF, an advised fund of the Silicon Valley Community Foundation (M.J.K.), the National Institute On Aging of the National Institutes of Health (R01AG062517 B.N.D, P30AG10129 C.D., and P30AG072972 C.D.), the University of California Office of the President (MRI-19-599956 B.N.D.), and the California Department of Public Health (CDPH; 19-10611 B.N.D.) with partial funding from the 2019 California Budget Act. We thank UC Davis, UCLA, and UC Irvine under CDPH grant #19-10611 for the validation dataset (released, see "Data availability"). We thank Drs. Julia K. Kofler and Charles L. White III for contributions to the training dataset (released in Wong et al.[13]). The views and opinions expressed in this article are those of the author and do not necessarily reflect the official policy or position of any public health agency of California or of the US government.

## Author contributions

Conceptualization: D.R.W., B.N.D., and M.J.K.; methodology: D.R.W. and M.J.K.; software: D.R.W.; validation: D.R.W., H.V.V., W.H.Y., S.D.M., and B.N.D.; formal analysis: D.R.W.; investigation: D.R.W., H.V.V., W.H.Y., S.D.M., B.N.D., A.C.M., and C.K.W.; resources: E.S.M., H.V.V., S.D.M., C.D., B.N.D., and M.J.K.; data curation: H.V.V., W.H.Y., S.D.M., C.K., J.P.G., B.N.D., and D.R.W.; writing—original draft: D.R.W.; writing—review & editing: D.R.W., B.N.D., and M.J.K; visualization: D.R.W.; supervision: B.N.D. and M.J.K.; project administration: D.R.W. and M.J.K.; funding acquisition: B.N.D. and M.J.K.

## Competing interests

The authors declare no competing interests.
