## [Peer Review File · Communications Biology]

Reviewers' comments:

Reviewer #1 (Remarks to the Author):

In this work, Wong et al. developed a deep learning model based on YOLOv3 for the quantification of A β pathologies in WSIs of Alzheimer's disease patients. The model was trained on a dataset of 659 images derived from a previous study and could quickly detect and score cored plaques and CAA. The authors showed that the model could achieve human expert-level accuracy, is scalable and reproducible, and does not require a GPU to run. This is an interesting approach to developing deep learning models to assess immunohistochemistry images, but there are several points that need to be addressed.

1. Semiquantitative evaluation of NFT and amyloid plaques make the pathological diagnosis of Alzheimer's disease. Please mention how quantitative evaluation of cored plaques and CAA, as was done in this study, would benefit the pathological diagnosis and patient care.
2. This study focused on cored plaques and CAA, but it is unclear why diffuse plaques were excluded. Please add a justification for this.
3. This study used immunostained slides with four different amyloid-beta antibodies. Pathologists can also find cored plaques and CAA on H&E stained slides. Can the model developed in this study also be applied to H&E slides to detect these pathologies?
4. The latest version of YOLO is version 7, but this study uses version 3. Please explain why the authors used an older version.
5. How does CERAD-like category scoring used in this study differ from CERAD scores? A brief explanation in the methods section would help the reader understand.
6. Please add more explanation about the WSIs used in this study: what is the thickness of the FFPE sections in the present study? Does this vary among the institutions? Did the counterstaining conditions vary among the institutions? Four different antibodies for amyloid- β were used, and according to Figure 1, the chromogenic dyes also differ from institution to institution. Please briefly explain these methods in detail.
7. It is mentioned that the WSIs used in this study were used in previous studies by the authors. However, it would be easier for the reader to understand if minimal information is also provided in this paper. Even the fact that these WSIs are brain sections from Alzheimer's disease patients is not mentioned in the Methods section. At least the age and sex of the patients and their APOE status should be included.
8. In the description of Fig. 1, it says 16 example images, but this reviewer found only 12. Also, please specify what antibody each image was stained with.
9. The data shown in Table 1 is difficult to read. This information should be included in the text. For example, the sentence "YOLOv3 model was 33 times faster than the older..." does not include the specific time taken, which is shown in Table 1. This reviewer thinks this information should be included in the text, not separately shown in a table.

Other minor points.

1. The use of abbreviations is not consistent.
2. Some references are not listed properly (truncated information), which makes it difficult to access the literature.

Reviewer #2 (Remarks to the Author):

The manuscript is written by a very well-known expert group, has a great structure, and explains different sections very well. The core idea is to rapidly create training datasets for detection of cored plaques and CAA to save the experts' time while achieving the same level of performance. The statistical analysis is appropriate, and the authors have provided the model configuration so the work is reproducible. The overall approach is interesting and creates thinking in the community when access to human-labeled data is not possible

However, my main biggest concern is the dependence on model version 2. It looks like that the first model based on generated candidates with human scoring would not be enough to generate a high-performing model. The CNN model was necessary here to weed out bad candidates, which was trained based on human annotations. This part needs more clarification. Specifically, the valid set could further be divided into a separate test set to report the performance of Model 1, Model 2, and the CNN-only model. At this point it's not clear whether taking the pre-trained CNN model's predictions would result in the same performance in the final fine-grained test-set.

Also, the third point of merit claim in discussion to have a model without GPU requirements is a bit too strong as the model architecture was used off-the-shelf.

Several things could be clarified in this manuscript:

- 1- How is the consensus-of-two achieved if 2 annotators say plaque, and 2 say none?
- 2- Diffuse pathologies were discarded from the dataset but no explanation is provided.
- 3- I don't believe claiming a strong correspondence between deposit counts and CERAD-like score is correct here given that the p-value was around 0.07.

Thanks to authors for the interesting work.

Response to Reviewer Comments

Reviewer 1

In this work, Wong et al. developed a deep learning model based on YOLOv3 for the quantification of A β pathologies in WSIs of Alzheimer's disease patients. The model was trained on a dataset of 659 images derived from a previous study and could quickly detect and score cored plaques and CAA. The authors showed that the model could achieve human expert-level accuracy, is scalable and reproducible, and does not require a GPU to run. This is an interesting approach to developing deep learning models to assess immunohistochemistry images, but there are several points that need to be addressed.

We thank the Reviewer for their support of the study's interest and guidance improving the manuscript.

1. Semiquantitative evaluation of NFT and amyloid plaques make the pathological diagnosis of Alzheimer's disease. Please mention how quantitative evaluation of cored plaques and CAA, as was done in this study, would benefit the pathological diagnosis and patient care.

Per the Reviewer's request, we have added clarification to the introduction motivating the need for quantitative evaluation in benefiting pathological diagnosis:

"There is a great need for quantitative, scalable means of assessing neuropathologies as current practices can suffer from interrater reliability issues and limited statistical analysis power, especially when relying on semiquantitative scores. Current neuropathology assessments for AD pathologies such as NFT stage (Alafuzoff et al. 2008; Braak and Braak 1991) and amyloid plaque phase (Thal et al. 2002) have added immensely to understanding the pathological progression of neurodegenerative diseases. Most schematics have used ordinal scales or the presence or absence of pathologies in specific areas. Recently, quantitative neuroanatomic-specific data have aided more robust correlations and understanding of selective vulnerability. This has led to further precision medicine approaches and the emergence of disease subtypes (Shakir and Dugger 2022; Murray et al. 2011; Montine and Montine 2015; Dugger et al. 2012)."

2. This study focused on cored plaques and CAA, but it is unclear why diffuse plaques were excluded. Please add a justification for this.

As diffuse plaques are not well-defined as objects, they are out of scope for this object detection method and study by definition. However, in future works, we aim to provide more scalable means for diffuse plaques and other neurodegenerative disease neuropathologies. We have added this justification to the Methods: "Diffuse plaques are not well-defined as objects and are thus out of scope for an object-detection method."

3. This study used immunostained slides with four different amyloid-beta antibodies. Pathologists can also find cored plaques and CAA on H&E stained slides. Can the model developed in this study also be applied to H&E slides to detect these pathologies?

The Reviewer proposes an exciting new direction, but it is outside the scope of this study, which focuses on IHC-stained whole slide images. This approach may indeed succeed on H&E WSIs, but it would require substantial human-annotator effort to collect the tens of thousands of annotations necessary for model training and assessment. The collection of human annotation datasets for IHCs were major contributions themselves as openly-released public resources in our prior studies (e.g., Tang et al, 2019; Wong et al, 2022). We indeed leveraged these data resources, some containing hundred of thousands of annotations by this point, in training the models for this study. We have added this as an accepted limitation in the Discussion section: "We confined our study to slides immunohistochemically stained with antibodies against A β . We have not determined the model's efficacy on other stain types, like hematoxylin and eosin (H&E) or silver stains."

4. The latest version of YOLO is version 7, but this study uses version 3. Please explain why the authors used an older version.

When we implemented this study, YOLOv3 was readily accessible and enjoyed the most community support. The YOLO development rate has been at least one new version per year, and we expect newer models than v7 will come out soon. This study's design, methods, and training are not specific to v3 or the YOLO architecture. Even though YOLOv3 is no longer the latest model, it rivals expert human annotators in other use cases, which was our goal. Indeed, many object detector architectures exist outside of the YOLO family of models. Comparing many such models was outside scope for this study.

5. How does CERAD-like category scoring used in this study differ from CERAD scores? A brief explanation in the methods section would help the reader understand.

The original CERAD scoring (Mirra 1991) assessed the densities of neuritic plaques per 1mm² area for Thioflavin-S and silver stains. As we evaluate plaque types on tissue immunohistochemically stained for A β instead, we used the term CERAD-like, as we and others have done in other papers.

6. Please add more explanation about the WSIs used in this study: what is the thickness of the FFPE sections in the present study? Does this vary among the institutions? Did the counterstaining conditions vary among the institutions? Four different antibodies for amyloid- β were used, and according to Figure 1, the chromogenic dyes also differ from institution to institution. Please briefly explain these methods in detail.

We have added information on slide thickness, staining, and chromogenic dyes to the Methods. For example, "UC Davis used the chromagen 3, 3'-diaminobenzidine and counterstained with hematoxylin. University of Pittsburgh used the chromagen Nova Red from Vector (Sk-4800). UT Southwestern used the Leica Bond robotic immunostaining platform with proprietary detection reagents. The detection kit used a red chromogen which employs an alkaline phosphatase enzyme and a Fast Red chromogenic substrate [...] UC Davis and UCLA had an approximate tissue thickness for each WSI of 6 μ m, while UCI had a tissue thickness of approximately 5 μ m. UC Davis, UCI, and UCLA all used the chromagen diaminobenzidine."

7. It is mentioned that the WSIs used in this study were used in previous studies by the authors. However, it would be easier for the reader to understand if minimal information is also provided in this paper. Even the fact that these WSIs are brain sections from Alzheimer's disease patients is not mentioned in the Methods section. At least the age and sex of the patients and their APOE status should be included.

We have clarified the Methods section to specify the use of brain section WSIs. We have added APOE status to Supplemental File 1, which includes Gender, Age, Race, Ethnicity, etc. Please also see Supplemental Figure 6 for a diagram showing data provenance.

8. In the description of Fig. 1, it says 16 example images, but this reviewer found only 12. Also, please specify what antibody each image was stained with.

We have fixed this typo and added antibody stains to the Figure 1 legend.

9. The data shown in Table 1 is difficult to read. This information should be included in the text. For example, the sentence "YOLOv3 model was 33 times faster than the older..." does not include the specific time taken, which is shown in Table 1. This reviewer thinks this information should be included in the text, not separately shown in a table.

We have removed Table 1 and included the information within the text instead. We added computing times as well: "Even without a GPU, this YOLOv3 model was 33 times faster (5 minutes 30 seconds / WSI) than the older GPU-enabled sliding window approach (181 minutes 30 seconds / WSI)."

Other minor points.

1. The use of abbreviations is not consistent.

2. Some references are not listed properly (truncated information), which makes it difficult to access the literature.

We have endeavored to correct both categories of these minor typographical deficiencies.

Reviewer 2

The manuscript is written by a very well-known expert group, has a great structure, and explains different sections very well. The core idea is to rapidly create training datasets for detection of cored plaques and CAA to save the experts' time while achieving the same level of performance. The statistical analysis is appropriate, and the authors have provided the model configuration so the work is reproducible. The overall approach is interesting and creates thinking in the community when access to human-labeled data is not possible.

We thank the reviewer for their kind words and support of the study and the manuscript.

However, my main biggest concern is the dependence on model version 2. It looks like that the first model based on generated candidates with human scoring would not be enough to generate a high-performing model.

*We found the Reviewer's point intriguing and thank them for raising it. This led us to evaluate model version 1 more thoroughly. We have added model version 1's performance on the held-out test set (as a new **Supplemental Figure 4**). Intriguingly, model version 1 also achieves human-expert-level performance, although it still does not perform as well as model 2, which we retain as the main focus.*

The CNN model was necessary here to weed out bad candidates, which was trained based on human annotations. This part needs more clarification.

We have clarified this filter step: "Secondly, we used our previously released consensus-of-two convolutional neural network (CNN) model to filter out low-quality CAA detections by removing detections that did not meet a final CNN classification prediction of 0.5 or higher".

Specifically, the valid set could further be divided into a separate test set to report the performance of Model 1, Model 2, and the CNN-only model.

We had already kept the test set separate from all models and collected it prospectively later in the study. Accordingly, we have added a new analysis evaluating Model-1 against this prospective test set (Supplemental Figure 4). However, comparing this study's YOLO models against the prior study's CNN model is not feasible as these models have incompatible tasks, and their outputs are not comparable: object identification vs. multiclass classification.

At this point it's not clear whether taking the pre-trained CNN model's predictions would result in the same performance in the final fine-grained test-set.

These models are not comparable – The CNN is a classifier that operates on predefined image tiles, whereas YOLO (Models 1 and 2) is an object detector that predicts boxes around objects of interest within a larger field of view. The CNN model, by construction, cannot rapidly localize objects; its closest approximation would be to laboriously "slide" its tile-sized view, pixel by pixel, across the entire gigapixel WSI, to construct an approximate heatmap, as we have done in prior studies.

Also, the third point of merit claim in discussion to have a model without GPU requirements is a bit too strong as the model architecture was used off-the-shelf.

The model was off the shelf, but inference does not require a GPU, and we believe this important characteristic makes the method more accessible for broad use. However, we clarified the claim, lest readers mistakenly think we are responsible for this non-GPU capability, as follows: "It is important to note that GPU dependency or lack thereof is dependent on model architecture."

Several things could be clarified in this manuscript:

1- How is the consensus-of-two achieved if 2 annotators say plaque, and 2 say none?

The consensus-of-two strategy derives from our previous paper (Wong et al, 2022). Under this strategy, if at least two annotators say plaque, it is considered a plaque, even if all other annotators say not. We have added a clarifying example in the Methods section: “For example, a consensus-of-two strategy means that at least two annotators identified the pathology, even if the remaining annotators did not.”

2- Diffuse pathologies were discarded from the dataset but no explanation is provided.

We have clarified this in the Methods section in response to Reviewer 1’s similar question 2.

3- I don’t believe claiming a strong correspondence between deposit counts and CERAD-like score is correct here given that the p-value was around 0.07.

We do not expect exact correlation between archival human CERAD-like annotations and model scoring. For instance, human annotations are generally calculated solely by assessing a single sub-region of interest within the broader WSI, whereas the model scores the entire tissue exhaustively. We have revised the abstract to clarify “...predictions that approximately correlated...” to ensure the reader does not conclude that we claim or seek a strong correspondence. Furthermore, we have flagged the one case where $p > 0.05$ in the Results so that the reader is aware: “We found significant differences for all category pairs at an alpha of 0.05, except for the pair “none” and “sparse” (student’s t-test p-value = 0.30, power = 0.62). Power for all other comparisons exceeded 0.99. ”

Thanks to authors for the interesting work.

REVIEWERS' COMMENTS:

Reviewer #1 (Remarks to the Author):

Several undefined abbreviations are used in the introduction, please correct this. Otherwise, the manuscript has been properly corrected.

Reviewer #2 (Remarks to the Author):

The authors have made remarkable improvements to the paper and have demonstrably tried to properly address comments and concerns raised during the original review. I would recommend this paper for publication in this journal.